# Deriving 3-D Surface Deformation Time Series with Strain Model and Kalman Filter from GNSS and InSAR Data

**Panfeng Ji** [1,2,3], **Xiaolei Lv** [1,2,*] and **Rui Wang** [1,2,3]

1 Key Laboratory of Technology in Geo-Spatial Information Processing and Application System, Aerospace Information Research Institute, Chinese Academy of Sciences, Beijing 100190, China; jipanfeng17@mails.ucas.ac.cn (P.J.); wangrui185@mails.ucas.ac.cn (R.W.)
2 Aerospace Information Research Institute, Chinese Academy of Sciences, Beijing 100094, China
3 School of Electronic, Electrical and Communication Engineering, University of Chinese Academy of Sciences, Beijing 100049, China
* Correspondence: academism2017@sina.com or lvxl@aircas.ac.cn

**Abstract:** This study proposes a new set of processing procedures based on the strain model and the Kalman filter (SM-Kalman) to obtain high-precision three-dimensional surface deformation time series from interferometric synthetic aperture radar (InSAR) and global navigation satellite system (GNSS) data. Implementing the Kalman filter requires the establishment of state and observation equations. In the time domain, the state equation is generated by fitting the pre-existing deformation time series based on a deformation model containing linear and seasonal terms. In the space domain, the observation equation is established with the assistance of the strain model to realize the spatial combination of InSAR and GNSS observation data at each moment. Benefiting from the application of the Kalman filter, InSAR and GNSS data at different moments can be synchronized. The time and measurement update steps are performed dynamically to generate a 3-D deformation time series with high precision and a high resolution in the temporal and spatial domains. Sentinel-1 SAR and GNSS datasets in the Los Angeles area are used to verify the effectiveness of the proposed method. The datasets include twenty-seven ascending track SAR images, thirty-four descending track SAR images and the daily time series of forty-eight GNSS stations from January 2016 to November 2018. The experimental result demonstrates that the proposed SM-Kalman method can produce high-precision deformation results at the millimeter level and provide two types of 3-D deformation time series with the same temporal resolution as InSAR or GNSS observations according to the needs of users. The new method achieves a high degree of temporal and spatial fusion of GNSS and InSAR data.

**Keywords:** GNSS; InSAR; 3-D deformation; strain model; Kalman filter





## 1. Introduction

Geological disasters, including natural disasters, such as earthquakes, land subsidence, volcanic movement and landslides and artificial disasters, such as engineering excavation, groundwater extraction and blasting, seriously affect human life. In recent decades, the interferometric synthetic aperture radar (InSAR) and global navigation satellite system (GNSS) have been widely used to detect surface deformation [1–3]. They can help humans understand the mechanism of the Earth's movement and monitor geological disasters.

These two Earth observation technologies, GNSS and InSAR, are complementary and suitable for combination. The existing continuous GNSS measurements can provide nearly time-continuous and high-precision three-dimensional surface positioning information [4]. However, limited by the conditions of the ground receiving equipment, GNSS often has a very low spatial resolution [5]. On the other hand, InSAR can measure large-scale surface displacements with high spatial resolution and high accuracy [6]. However, InSAR can only provide deformation measured along the line of sight (LOS) between the Earth's surface and the radar.

The displacements in the LOS direction are the projection of the real 3-D (east–west, north–south and up–down) displacements. Only relying on the monitoring in the LOS direction may lead to the misjudgment of the real surface information [7]. In recent decades, scholars have conducted a large amount of research on combining the two complementary ground observation techniques, GNSS and InSAR, to obtain high-precision 3-D deformation fields. The research results largely fall into three categories. The first type of method is represented by the analytical optimization method [8,9], the BFGS method [10], the ant colony optimization method [11] and the MRF-L1 regularization method [12].

The basic principle of these methods is to interpolate the deformation of discrete GNSS points into the same spatial grid of InSAR and then solve the 3-D deformation by establishing an optimization model based on Markov random field and Bayesian estimation theory. The second type of method is represented by the Helmert VCE method [13], the BQUE (best quadratic unbiased estimator) method [14] and the IAUE (iterated almost unbiased estimation) method [15]. The core idea of this type of method is to estimate the posterior variance components of InSAR and GNSS based on the theory of geodesy and then solve the 3-D deformation through the principle of weighted least squares.

The third type of method is typically represented by the SISTEM (Simultaneous and Integrated Strain Tensor Estimation From Geodetic and Satellite Deformation Measurements) method [16]. Based on the strain model in structural geology [17], it can directly combine the InSAR and GNSS data to solve 3-D deformation without the pre-interpolation step of GNSS. In addition, statistical machine-learning methods have also been applied to estimate 3-D deformation by fusing both GNSS and InSAR data [18].

Previous studies have obtained the 3-D surface displacement field; however, they have needed to downsample the GNSS time series to the corresponding InSAR deformation observation moments before fusion. Moreover, they have only focused on the displacement extracted from the two-scene SAR images or the deformation rate from the multi-scene SAR images. They have neither taken full advantage of the large time resolution of GNSS nor have they been able to extract the 3-D surface deformation time series to analyze the time evolution of deformation intuitively. In recent years, some attempts have been made to extract 3-D deformation time series based on multi-orbit InSAR data or geological models [19–21].

In 2013, Hu et al. proposed a method for extracting 3-D deformation time series from multi-track InSAR data based on the Kalman filter [22]. The SBAS-based multidimensional small baseline subset (MSBAS) method was then proposed to acquire 2-D or 3-D deformation time series from multiple InSAR data [23]. In addition, Liu et al. proposed the KFInSAR (Kalman filter InSAR) method based on the strain model and Kalman filter to obtain 3-D deformation time series from multiple InSAR data [24].

The strain model is based on elasticity theory and can be used to model stress–strain relationships between points in a geographic space [25]. The strain model introduced in KFInSAR can reduce the high-frequency noises of InSAR, such as atmospheric errors. However, due to SAR satellites' nearly north–south flight direction, it is difficult to extract high-precision north–south deformation time series based on multi-orbit InSAR data. The above methods can only estimate reliable 2-D deformation time series in the east–west and vertical directions. GNSS technology can observe very high-precision displacement in the north–south direction. Its fusion with InSAR can largely overcome the shortcoming mentioned above. However, there are few studies on combining the two kinds of data to extract 3-D deformation time series.

In this paper, we present a methodology based on the strain model and the Kalman filter (SM-Kalman) that allows the direct integration of various InSAR and GNSS datasets to produce a 3-D deformation time series when InSAR and GNSS data sets with overlapping temporal and spatial coverage are available. This allows the integration of InSAR data with different wave-band, azimuth and incidence angles, as well as different spatial and temporal sampling and resolution, including airborne and spaceborne data from sensors

with varying parameters. In KFInSAR, which was proposed by Liu et al., the strain model is used to attenuate the noises of InSAR.

In SM-Kalman, this model is used to build the observation equation to combine GNSS and InSAR observations at each moment in the spatial domain. In addition, a deformation model containing linear and seasonal terms is used to establish the state equation based on a pre-existing deformation time series. Then, the 3-D deformation time series are estimated moment by moment through Kalman filtering steps, such as time and measurement updates. Sentinel-1 SAR and GNSS datasets in the Los Angeles area are used to verify the effectiveness of the proposed method. The datasets include twenty-seven ascending track SAR images, thirty-four descending track SAR images and the daily time series of forty-eight GNSS stations from January 2016 to November 2018.

## 2. Methods

### 2.1. Correction of InSAR Observations

GNSS can provide high-precision positioning services because many errors,, such as ionospheric delays, tropospheric delays and clock offsets, are mitigated in its design [26]. However, the accuracy of InSAR is degraded by unmodeled atmospheric delays, satellite orbit errors and errors in interferometric processing, etc. Additionally, InSAR is a relative observation technique, which means that the estimated displacements or deformation rates are relative to a reference point [27].

In this paper, the Persistent Scatterer Interferometry (PSI) technique [28] is applied to process the ascending and descending orbit SAR datasets. The PSI technique focuses on the persistent scatterer (PS) points that exhibit relatively constant scattering properties over time, significantly reducing the effects of geometric decorrelation. Due to the use of a radar acquisition stack, atmospheric phase components from observations can be estimated and removed [29]. However, PSI still only yields relative observational information. Therefore, the unification of the two reference frames must be completed before implementing the fusion of GNSS and InSAR.

Traditionally, the method of unifying the results of InSAR and GNSS data is to correct the GNSS displacements to the relative reference frame as InSAR. When this method is adopted, a certain GNSS point with small displacements that is far away from the large deformation area, is selected as a common reference point for InSAR and GNSS [12]. This method is generally applied in the combination of DInSAR and GNSS to monitor large deformation scenarios, such as earthquakes, volcanoes and landslides. However, when the deformation distribution in the entire SAR image is relatively uniform, the method is no longer applicable [18].

Generally speaking, there is a systematic deviation between the relative InSAR and the actual absolute measurement. It can be compensated with external data. In this paper, a method of polynomial fitting is used to correct InSAR observations using GNSS data [14,27]. The specific operation steps are as follows:

1.　Project the GNSS 3-D displacements corresponding to the InSAR acquisition moments to the LOS direction

$$d_{gnss}^{los} = [s_e \ s_n \ s_u][d_{ge} \ d_{gn} \ d_{gu}]^T \tag{1}$$

where $d_{ge}$, $d_{gn}$ and $d_{gu}$ are the GNSS displacements in the east–west, north–south and up–down directions, respectively. The unit project vector $[s_e \ s_n \ s_u] = [-sin\theta sin(\alpha - 3/2\pi) \ -sin\theta cos(\alpha - 3/2\pi) \ cos\theta]$. $\theta$ denotes the incidence angle of the satellite site and $\alpha$ represents the heading angle, that is, the clockwise angle between the north and the flight of the satellite. Through Equation (1), the 3-D GNSS displacements are projected to the LOS direction $d_{gnss}^{los}$.

2.  The polynomial fitting can establish the relationship between GNSS projection values and InSAR observations at the same location and moment. Assuming a second-order polynomial fit, the relationship between the two observations can be expressed as

$$d_{gnss}^{los} = a \cdot d_{los}^2 + b \cdot d_{los} + c \tag{2}$$

where $a$, $b$ and $c$ are the fitting coefficients, and $d_{los}$ is the InSAR observation in the LOS direction. It can be seen that as long as the number of GNSS points in the study area is greater than three, the optimal polynomial coefficients can be obtained through the least squares method. Of course, the more GNSS points involved in the fitting, the more accurate the coefficients will be.

3.  Correct the InSAR observations for each pixel using the polynomial coefficients obtained. The above steps will be performed independently for each observation moment of the ascending and descending track InSAR datasets.

### 2.2. State Equation

The Kalman filter is a dynamic data processing method that considers the state and relevance of the data in the time domain. Since there is no need to store a large amount of historical data, it constantly predicts and revises the data during processing [30]. The state and observation equations must first be established before implementing the Kalman filter. In the time domain, the displacement of the Earth's surface is a time-dependent evolution process, except for in the case of sudden events, such as earthquakes and volcanoes [31]. In the spatial domain, the surface deformation of a point is also related to the surrounding points [32]. Together, these two facts form the theoretical basis for the establishment of the state and observation equations.

It is assumed that the deformation time series consists of an initial term, a linear term and multiple periodic terms. If the influence of noises is not considered first, the deformation time series can be expressed as [33]

$$d(t) = a_0 + a_1 t + a_2 cos(\frac{2\pi}{T} * t) + a_3 sin(\frac{2\pi}{T} * t) \tag{3}$$

where $t$ is the epoch relative to the initial moment, $a_0$ is the initial deformation, $a_1$ is the linear deformation rate, and $a_2$ and $a_3$ are the coefficients of the seasonal items, respectively. $T$ is the period of seasonal deformation. If $T = 1$, it is an annual signal, and if $T = 0.5$, it is a semi-annual signal [34]. If Equation (3) is discretized, the two deformation time series at $k$ and the previous moment $k - 1$ can be expressed as

$$d_{k-1} = a_0 + a_1 t_{k-1} + a_2 cos(\frac{2\pi}{T} * t_{k-1}) + a_3 sin(\frac{2\pi}{T} * t_{k-1}) \tag{4}$$

$$d_k = a_0 + a_1(t_{k-1} + \delta t) + a_2 cos(\frac{2\pi}{T} * (t_{k-1} + \delta t)) + a_3 sin(\frac{2\pi}{T} * (t_{k-1} + \delta t)) \tag{5}$$

where $\delta t$ represents the time interval between moment $k$ and $k - 1$. By subtracting and shifting the terms of Equations (4) and (5), we can obtain the relationship between the displacements at two moments

$$d_k = d_{k-1} + a_1 \delta t + a_2[cos(\frac{2\pi}{T} * t_k) - cos(\frac{2\pi}{T} * t_{k-1})] + a_3[sin(\frac{2\pi}{T} * t_k) - sin(\frac{2\pi}{T} * t_{k-1})] \tag{6}$$

Then, we extend the above equation to the deformation in the east, north and up directions. The state equation can be generated by writing Equation (6) in matrix form.

$$\mathbf{x}_k = \mathbf{F}_{k-1}\mathbf{x}_{k-1} + \mathbf{A}_{k-1}\mathbf{u}_{k-1} + \mathbf{w}_{k-1} \tag{7}$$

where the state $\mathbf{x}_k = [d_{k,e}\ d_{k,n}\ d_{k,u}]^T$ is the unknown 3-D deformation vector at $k$, $\mathbf{w}_{k-1}$ is the state noise vector, and $\mathbf{F}_{k-1}$ is a three-dimensional identity matrix

$$\mathbf{F}_{k-1} = \mathbf{I}_3 \tag{8}$$

$$\mathbf{u}_{k-1} = \begin{bmatrix} \delta t & cos(\frac{2\pi}{T} * t_k) - cos(\frac{2\pi}{T} * t_{k-1}) & sin(\frac{2\pi}{T} * t_k) - sin(\frac{2\pi}{T} * t_{k-1}) \end{bmatrix}^T \tag{9}$$

and

$$\mathbf{A}_{k-1} = \begin{bmatrix} a_e^1 & a_e^2 & a_e^3 \\ a_n^1 & a_n^2 & a_n^3 \\ a_u^1 & a_u^2 & a_u^3 \end{bmatrix} \tag{10}$$

Deformation time series generally exhibit obvious seasonal signals. If the anniversary term and the semi-annual term are both considered—that is, $T$ takes both 1 and 0.5, then the new $\mathbf{u}_{k-1}$ will become

$$\mathbf{u}_{k-1} = \begin{bmatrix} \delta t \\ cos(2\pi * t_k) - cos(2\pi * t_{k-1}) \\ sin(2\pi * t_k) - sin(2\pi * t_{k-1}) \\ cos(4\pi * t_k) - cos(4\pi * t_{k-1}) \\ sin(4\pi * t_k) - sin(4\pi * t_{k-1}) \end{bmatrix} \tag{11}$$

and

$$\mathbf{A}_{k-1} = \begin{bmatrix} a_e^1 & a_e^2 & a_e^3 & a_e^4 & a_e^5 \\ a_n^1 & a_n^2 & a_n^3 & a_n^4 & a_n^5 \\ a_u^1 & a_u^2 & a_u^3 & a_u^4 & a_u^5 \end{bmatrix} \tag{12}$$

Furthermore, other models, such as polynomial functions, can also be applied for the simulation of deformation time series. Assuming that a third-order polynomial is used for fitting, the deformation time series is

$$d(t) = a_0 + a_1 t + a_2 t^2 + a_3 t^3 \tag{13}$$

and

$$\mathbf{u}_{k-1} = \begin{bmatrix} \delta t & t_k^2 - t_{k-1}^2 & t_k^3 - t_{k-1}^3 \end{bmatrix}^T \tag{14}$$

Some other fitting functions, such as the cubic splines and the orthogonal polynomials, can also be used to construct the state equation. In any case, the establishment of the state equation is flexible. Users may choose an appropriate fitting function according to the actual deformation conditions of the study area, which can be easily prejudged by the time series of GNSS.

*2.3. Observation Equation*

In recent years, the strain model has received extensive attention in the application of GNSS and InSAR. There are three main categories of its applications: (1) It can be used to interpolate the 3-D deformation of discrete GNSS points and to calculate the crustal strain rates [35]. (2) Applying the SISTEM method based on the strain model, GNSS and InSAR data can be directly combined to extract 3-D deformation without the pre-interpolation of GNSS points [16]. (3) Based on the strain model, the pixels around the target point can be linked, creating a large number of redundant observations.

Therefore, the variances of the target point can be correctly estimated using the posterior variance component estimation method [15,36]. The advantage of the strain model is that it is based on the elasticity theory and can be used to model the relationship between the position gradient and displacement of the target point and surrounding pixels in the same geographic space. This makes it possible to combine the observations of the target point with the surrounding observations for a comprehensive analysis, rather than focusing on individual points as before [17]. In this paper, the strain model is adopted to construct the observation equation of SM-Kalman.

As shown in Figure 1, in a SAR image, it assumes that the 3-D coordinates and displacements of the target point $P_0$ are $\mathbf{c}_0 = [c_e^0 \; c_n^0 \; c_u^0]^T$ and $\mathbf{d}_0 = [d_e^0 \; d_n^0 \; d_u^0]^T$, respectively. Similarly, the 3-D coordinates and displacements of $N$ adjacent GNSS points are $\mathbf{c}_i = [c_{ge}^i \; c_{gn}^i \; c_{gu}^i]^T$ and $\mathbf{d}_i = [d_{ge}^i \; d_{gn}^i \; d_{gu}^i]^T (i = 1, 2, \cdots, N)$, respectively. Based on the elastic strain model theory, the position and displacement vectors of the target point $P_0$ and the $N$ GNSS points can be modeled as [37]

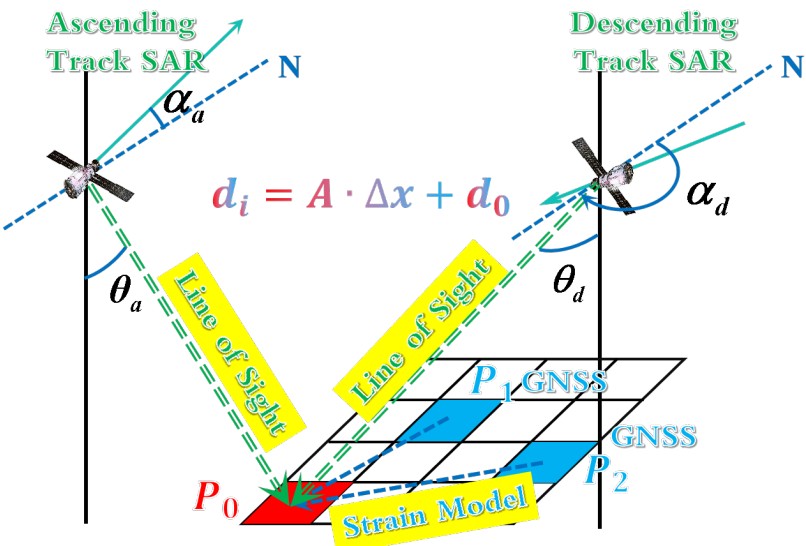

**Figure 1.** The strain model to fusion GNSS and InSAR measurements in the spatial domain.

$$\mathbf{d}_i = \mathbf{B}\Delta\mathbf{c}_i + \mathbf{d}_0, \; i = 1, 2, \cdots, N \tag{15}$$

where $\Delta\mathbf{c}_i = \mathbf{c}_i - \mathbf{c}_0$ represents the position difference vector from the $i$-th GNSS point $P_i$ to the target point $P_0$. $\mathbf{B}$ denotes the stain parameter matrix and can be divided into a strain tensor component $\mathbf{B}_s$ [38]

$$\mathbf{B}_s = \frac{1}{2}(B_{ij} + B_{ji})\mathbf{e}_i \otimes \mathbf{e}_j = \begin{bmatrix} \epsilon_{11} & \epsilon_{12} & \epsilon_{13} \\ \epsilon_{21} & \epsilon_{22} & \epsilon_{23} \\ \epsilon_{31} & \epsilon_{32} & \epsilon_{33} \end{bmatrix} \tag{16}$$

and a rigid body tensor component $\mathbf{B}_r$ [38]

$$\mathbf{B}_r = \frac{1}{2}(B_{ij} - B_{ji})\mathbf{e}_i \otimes \mathbf{e}_j = \begin{bmatrix} 0 & -\omega_3 & \omega_2 \\ \omega_3 & 0 & -\omega_1 \\ -\omega_2 & \omega_1 & 0 \end{bmatrix} \tag{17}$$

where $\epsilon_{\{\cdot\}}$ and $\omega_{\{\cdot\}}$ are the parameters of stain model. $\mathbf{e}_{\{\cdot\}}$ is the canonical base vector of the Cartesian reference system and $\otimes$ is the tensor product. In a compact form, Equation (15) can be written as

$$\mathbf{d}_i = \mathbf{H}_i\mathbf{x}, \; i = 1, 2, \cdots, N \tag{18}$$

where $\mathbf{H}_i$ is the strain model matrix which is given in reference [35]. The unknown vector is

$$\mathbf{x} = [d_e \; d_n \; d_u \; \epsilon_{11} \; \epsilon_{12} \; \epsilon_{13} \; \epsilon_{22} \; \epsilon_{23} \; \epsilon_{33} \; \omega_1 \; \omega_2 \; \omega_3]^T \tag{19}$$

Please note that only $N$ GNSS observations are included in $\mathbf{d}$. In order to fuse InSAR and GNSS points to estimate the 3-D deformation of $P_0$ in the space domain, the ascending and descending track InSAR observations of $P_0$ can be added to the vector $\mathbf{d}$. The new $\mathbf{d} = [d_{ge}^1 \; d_{gn}^1 \; d_{gu}^1 \; \cdots \; d_{ge}^N \; d_{gn}^N \; d_{gu}^N \; d_{los}^a \; d_{los}^d]^T$. Similarly, when considering the geometric relationship between the 3-D deformation in the east–west, north–south and up–down directions and the 1-D deformation in the LOS direction, as shown in Equation (1), the ma-

trix **H** can also be rewritten as shown. The superscripts $a$ and $d$ in the unit project vector $[s_e^a \ s_n^a \ s_u^a]$ and $[s_e^d \ s_n^d \ s_u^d]$ denote ascending and descending orbits, respectively.

$$\mathbf{H} = \begin{bmatrix} 1 & 0 & 0 & \Delta c_{ge}^1 & \Delta c_{gn}^1 & \Delta c_{gu}^1 & 0 & 0 & 0 & 0 & \Delta c_{gu}^1 & -\Delta c_{gn}^1 \\ 0 & 1 & 0 & 0 & \Delta c_{ge}^1 & 0 & \Delta c_{gn}^1 & \Delta c_{gu}^1 & 0 & -\Delta c_{gu}^1 & 0 & \Delta c_{ge}^1 \\ 0 & 0 & 1 & 0 & 0 & \Delta c_{ge}^1 & 0 & \Delta c_{gn}^1 & \Delta c_{gu}^1 & \Delta c_{gn}^1 & -\Delta c_{ge}^1 & 0 \\ \vdots & \vdots & \vdots & \vdots & \vdots & \vdots & \vdots & \vdots & \vdots & \vdots & \vdots & \vdots \\ 1 & 0 & 0 & \Delta c_{ge}^N & \Delta c_{gn}^N & \Delta c_{gu}^N & 0 & 0 & 0 & 0 & \Delta c_{gu}^N & -\Delta c_{gn}^N \\ 0 & 1 & 0 & 0 & \Delta c_{ge}^N & 0 & \Delta c_{gn}^N & \Delta c_{gu}^N & 0 & -\Delta c_{gu}^N & 0 & \Delta c_{ge}^N \\ 0 & 0 & 1 & 0 & 0 & \Delta c_{ge}^N & 0 & \Delta c_{gn}^N & \Delta c_{gu}^N & \Delta c_{gn}^N & -\Delta c_{ge}^N & 0 \\ s_e^a & s_n^a & s_u^a & 0 & 0 & 0 & 0 & 0 & 0 & 0 & 0 & 0 \\ s_e^d & s_n^d & s_u^d & 0 & 0 & 0 & 0 & 0 & 0 & 0 & 0 & 0 \end{bmatrix} \tag{20}$$

Based on Equation (18), the observation equation at moment $k$ can be expressed as

$$\mathbf{d}_k = \mathbf{H}_k \mathbf{x}_k + \mathbf{v}_k \tag{21}$$

where $\mathbf{v}_k$ is the noise vector of observations.

*2.4. SM-Kalman*

It can be seen that the state Equation (7) can be established by differentiating the deformation time series of two adjacent moments, and the observation Equation (21) can also be successfully established by introducing the strain model. Based on the two equations, the framework of SM-Kalman can be established. During the implementation of SM-Kalman, the state vector in the state equation should be consistent with the state vector in the observation equation. Therefore, Equation (7) needs to be extended to the dimension of Equation (19). Assuming that **w** and **v** are uncorrelated white noises, the statistical properties of the covariance matrix are

$$E(\mathbf{w}_k \mathbf{w}_j^T) = \mathbf{Q}\delta_{k-j}; \quad E(\mathbf{v}_k \mathbf{v}_j^T) = \mathbf{R}\delta_{k-j}; \quad E(\mathbf{w}_k \mathbf{v}_j^T) = \mathbf{0} \tag{22}$$

Figure 2 indicates the flowchart of operating the SM-Kalman to obtain the 3-D deformation time series. Since the establishment of the state Equation (7) needs to fit the 3-D time series, the SM-Kalman filter will start from $k = n$ instead of $k = 0$ ($n$ is set as the number of coefficients in the time series). It is recommended that the 3-D displacement time series from $k = 0$ to $k = n - 1$ are obtained using traditional SISTEM [16] or analytical optimization methods [9].

Before the Kalman filter is executed, the initial state vector and covariance matrix of the state vector need to be input. At $k = n$, the initial state vector and initial covariance matrix for the SM-Kalman filter are provided by the results of SISTEM at $k = n - 1$. Then, the time update (prediction) and measurement update (correction) can be performed according to the general steps of Kalman filtering. The time update is performed first. We predict the prior state $\hat{\mathbf{x}}_{k,k-1}$ and prior covariance matrix $\mathbf{D}_{\hat{\mathbf{x}}_{k,k-1}}$ of the current $k$ moment using the state vector and covariance matrix of the previous $k - 1$ moment [30]

$$\hat{\mathbf{x}}_{k,k-1} = \mathbf{F}_{k-1}\hat{\mathbf{x}}_{k-1} + \mathbf{A}_{k-1}\mathbf{u}_{k-1} \tag{23}$$

$$\mathbf{D}_{\hat{\mathbf{x}}_{k,k-1}} = \mathbf{F}_{k-1}\hat{\mathbf{x}}_{k-1} + \mathbf{F}_{k-1}^T + \mathbf{Q}_{k-1} \tag{24}$$

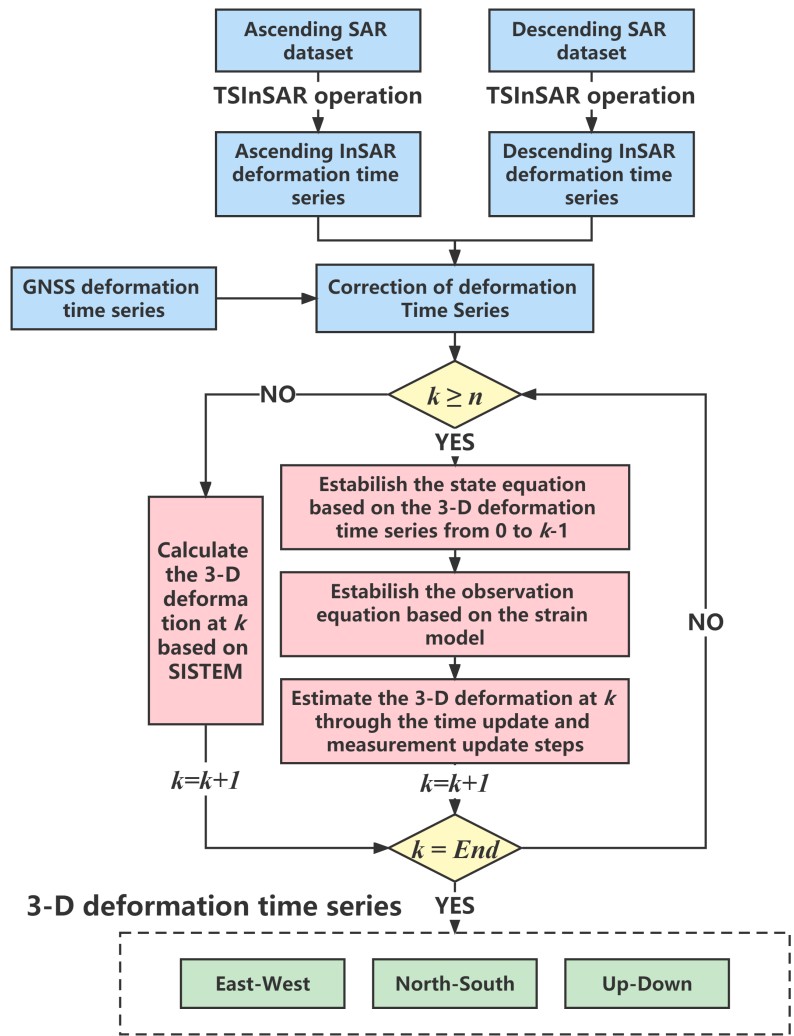

**Figure 2.** Flow of operating the SM-Kalman method.

The observations at the current moment $k$ can then be used to correct the prior state and prior covariance matrices obtained from the time update. This step is called the measurement update [30].

$$\mathbf{K}_k = \mathbf{D}_{\hat{\mathbf{x}}_{k,k-1}} \mathbf{H}_k'^T (\mathbf{H}_k' \mathbf{D}_{\hat{\mathbf{x}}_{k,k-1}} \mathbf{H}_k'^T + \mathbf{R}_k)^{-1} \tag{25}$$

$$\hat{\mathbf{x}}_k = \hat{\mathbf{x}}_{k,k-1} + \mathbf{K}_k (\mathbf{d}_k' - \hat{\mathbf{x}}_{k,k-1}) \tag{26}$$

$$\mathbf{D}_{\hat{\mathbf{x}}_k} = (\mathbf{I} - \mathbf{K}_k \mathbf{H}_k') \mathbf{D}_{\hat{\mathbf{x}}_{k,k-1}} \tag{27}$$

where $\mathbf{K}_k$ is the Kalman gain, $\hat{\mathbf{x}}_k$ and $\mathbf{D}_{\hat{\mathbf{x}}_k}$ are the posterior state vector and posterior covariance matrix, respectively. It should be noted here that $\mathbf{H}'$ and $\mathbf{d}'$ cannot be directly equivalent to $\mathbf{H}$ and $\mathbf{d}$ in Equation (21). They are expressed as follows

$$\mathbf{P}' = CD(\mathbf{P}) \tag{28}$$

$$\mathbf{H}' = \mathbf{P}' \mathbf{H} \tag{29}$$

$$\mathbf{d}' = \mathbf{P}' \mathbf{d} \tag{30}$$

where **P** is the weight matrix of the observation vector and is set as a diagonal matrix

$$\mathbf{P} = diag(P_{ge}^1 \; P_{gn}^1 \; P_{gu}^1 \; \cdots \; P_{ge}^N \; P_{gn}^N \; P_{gu}^N \; P_{los}^a \; d_{los}^d) \tag{31}$$

The weights of *N* GNSS points and InSAR are provided by the attenuation function of the distance and the sample semivariogram [39], respectively. *CD* represents the Cholesky decomposition of the matrix. The reason for transforming strain model matrix **H** and observation vector **d** through Equations (28)–(30) is that, for the target point, the weights of *N* GNSS observation points should vary with distance. The closer the GNSS points are, the larger the allocated weights should be; otherwise, smaller weights should be allocated. This effect should be reflected in the estimation of the posterior state vector.

Benefiting from the application of the Kalman filter, InSAR and GNSS data at different moments can be synchronized. At *k*, the three types of observations, i.e., GNSS, ascending track InSAR and descending track InSAR, may not all be available, and thus the observation vector $\mathbf{d}_k$ will change dynamically. That is to say that when there are only *N* GNSS observations at *k*, $\mathbf{d}_k$ is a $3 * N$ dimensional vector; when there is one InSAR observation, it is a $3 * N + 1$ dimensional vector; when there are InSAR observations in both ascending and descending orbits, it is a $3 * N + 2$ dimensional vector. As *k* continues to increase, SM-Kalman will be executed iteratively until the end moment, as shown in Figure 2.

The variance–covariance matrix **R** of the observation vector **d** determines the observation accuracy of the *N* GNSS points, ascending track and descending track InSAR. **R** should also be a diagonal matrix similar to Equation (31). Readers can feel the difference between **P** and **R** here. In this paper, the elements of $\mathbf{R}_k$ at the current moment *k* are set as the standard deviations provided by the GNSS positioning and PSI techniques. Inevitably, there are errors between the deformation predicted by the state update at *k* and the actual value.

They are measured by the variance–covariance matrix **Q**. In the processing of SM-Kalman, the 3-D deformation time series from 0 to $k - 1$ can be fitted on *N* GNSS points. Then, the fitted function is applied to predict the quasi-observed values at *N* GNSS points at the current moment *k*, and then the matrix **Q** can be filled with the standard deviations between these quasi-observed and the real observed values.

Theoretically, the 3-D deformation time series estimated using SM-Kalman can reach the observation time resolution of InSAR or the observation time resolution of GNSS. Users can choose the results of two temporal resolutions according to their needs. If the time resolution of GNSS is used, SM-Kalman will make full use of the high-resolution data of GNSS in the time domain to achieve a high degree of fusion of GNSS data and InSAR in the space-time domain.

## 3. Results

### 3.1. InSAR and GNSS Results

In Figure 3, the blue and green solid lines outline the coverage area of the ascending and descending track SAR data used in this experiment. The study area is located in Los Angeles, which is within the overlapping area of the ascending and descending orbit data. Sixty-one Sentinel-1 SAR acquisitions, including 27 ascending track images and 34 descending track images, over the area of Los Angeles from 9 January 2016 to 18 November 2018, were used in the experiment, as listed in Table 1. PSI technology was applied to process the ascending track and descending track SAR datasets in the Los Angeles area.

Figure 4 is a visual display of the space-time baseline of the interferometric pairs composed of the ascending and descending orbit data. It can be seen that the PSI technology is processed based on a single reference image (represented by a red star), and the composing interferometric pairs are star-shaped. When operating PSI, the selection of the reference image directly affects the extraction of subsequent deformation information.

Ferretti suggested that the selection of the reference image should be mainly based on the time baseline [28]. In this experiment, the reference images for the processing of ascending

and descending track interferometry were both dated to May 2017, near the center of the images stack. In this way, the influence of time decoherence can be better avoided.

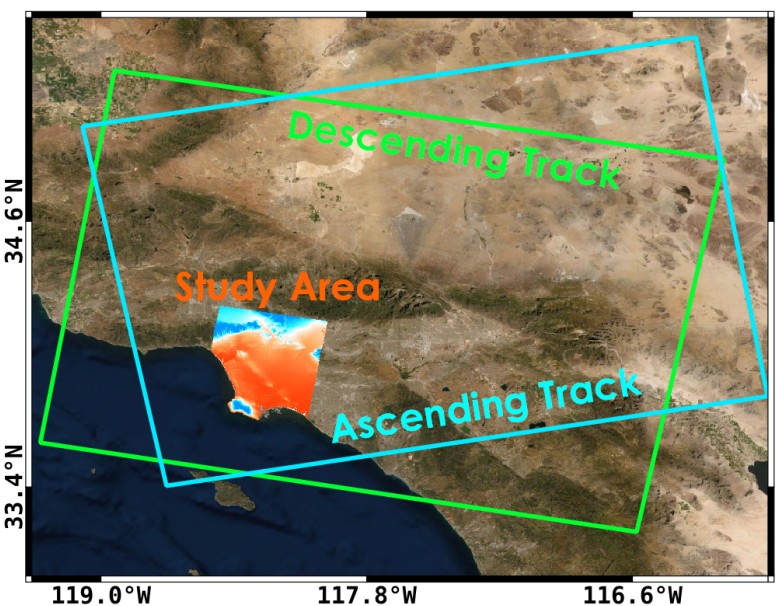

**Figure 3.** Coverage of the ascending track and descending track SAR data in the study area.

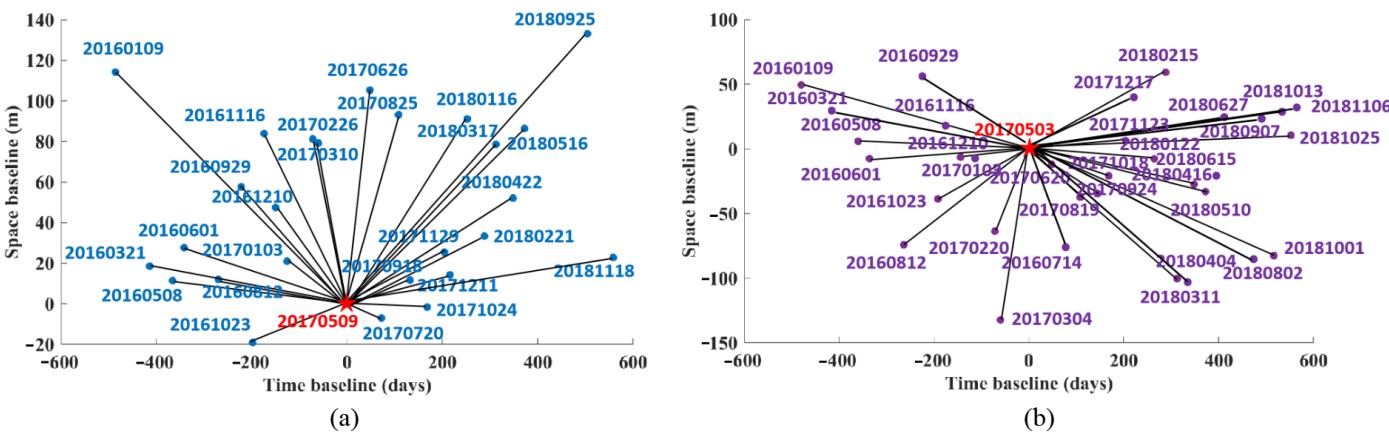

**Figure 4.** Temporal and spatial baselines of SAR datasets, (**a**) ascending track and (**b**) descending track.

A multi-looking operation with two looks in the range and ten looks in the azimuth directions was also performed to reduce the phase noises in the interferograms, followed by a filtering process with the improved Goldstein filter [40]. The AW3D digital elevation model produced by the Japan Aerospace Exploration Agency was employed to remove the topographic phase [41], which is shown in Figure 5. Furthermore, the interferometric phases were unwrapped using the minimum cost flow (MCF) algorithm for pixels whose coherence values exceeded 0.3 [42].

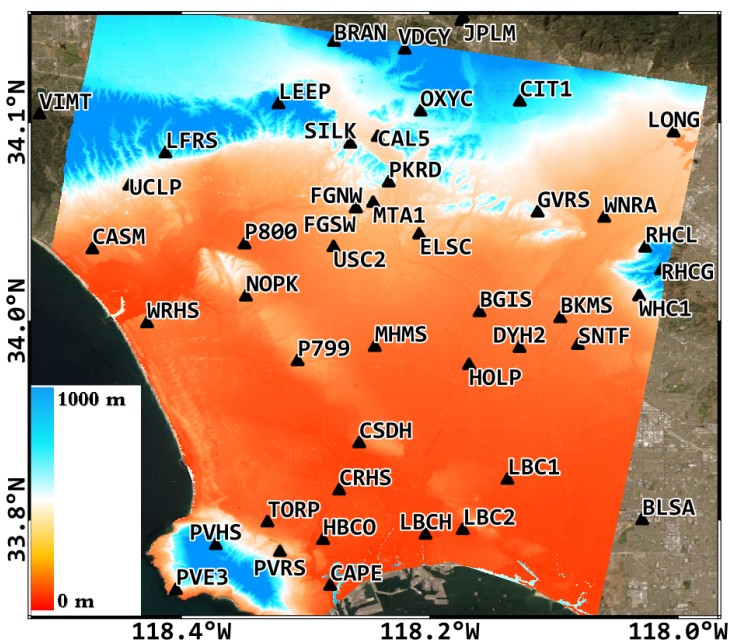

**Figure 5.** Topographic map of the study area. The black triangles represent the position of GNSS stations. The four characters next to each GNSS station represent its name.

**Table 1.** SAR data parameters used in the Los Angeles case.

| Track | Time Span | Number | Incidence Angle | HEADING |
|---|---|---|---|---|
| Ascending Track | 20160109–20181118 | 27 | 33.985 | −12.948 |
| Descending Track | 20160109–20181106 | 34 | 39.271 | −170.105 |

Figure 5 shows the distribution of 48 GNSS stations in the Los Angeles area in this experiment. The daily coordinates of the GNSS stations were in the IGS14 Earth reference frame and were downloaded from the Nevada Geodetic Laboratory. They were used to correct the geocoded InSAR measurements and aid the InSAR results to obtain the derived 3-D ground displacement time series.

Figures 6 and 7 show the deformation time series of the ascending and descending InSAR, respectively. The length of each arrow reflects the displacements of the GNSS station at the corresponding moment. For the convenience of presentation, eight results from the ascending track and descending track InSAR deformation time series were selected. The deformation time series of both ascending and descending orbits identified some evident deformation regions. However, due to the limitation, InSAR can only monitor the LOS-directed deformation. Moreover, the area has been experiencing complex ground motion due to a combined effect of tectonic motion and anthropogenic activities [43–45], InSAR measurements cannot fully reflect the actual ground motion in a comprehensive manner.

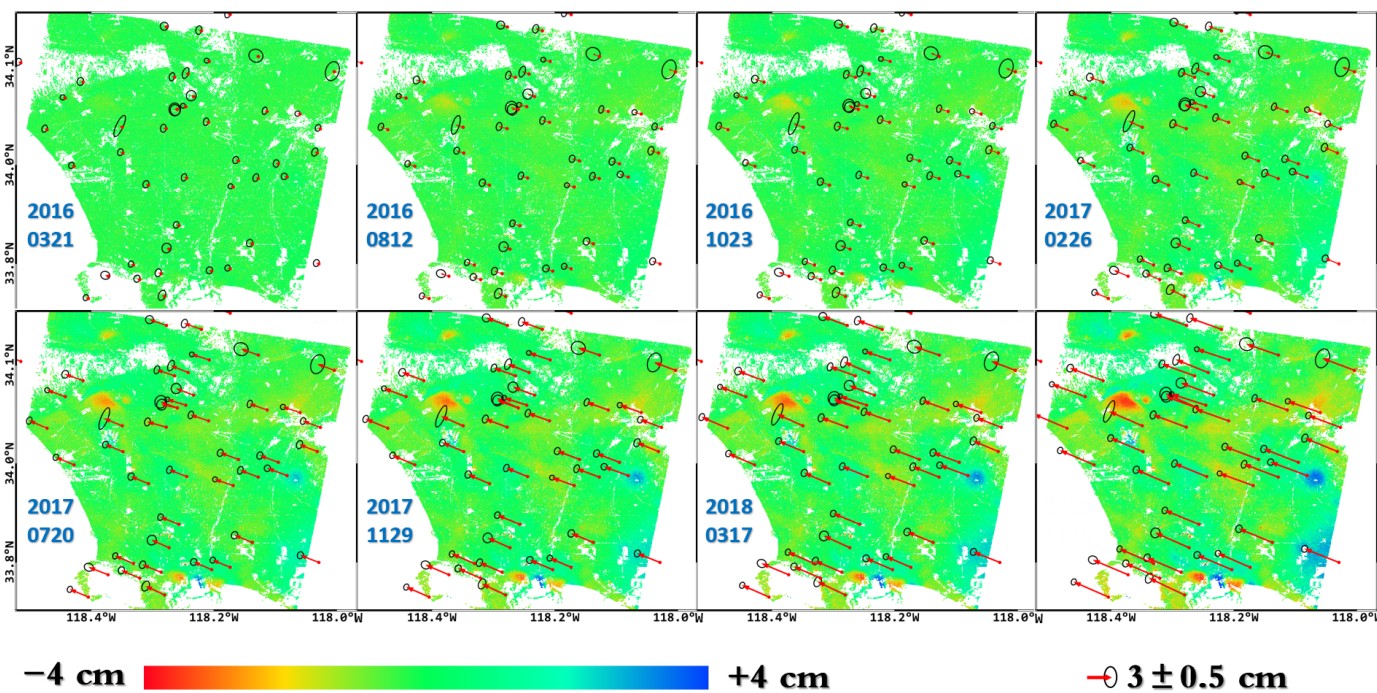

**Figure 6.** Deformation time series maps of ascending InSAR. GNSS sites with displacement vectors are indicated by red arrows. The black ellipses represent the horizontal errors.

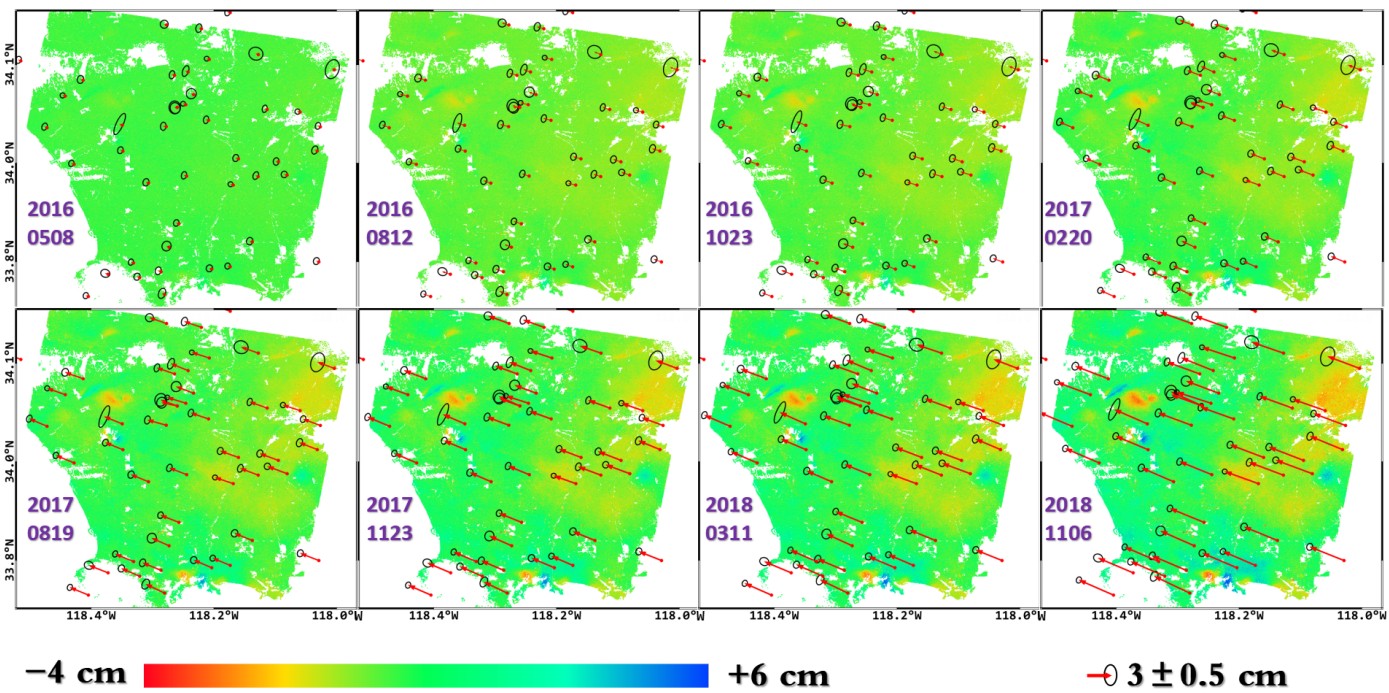

**Figure 7.** Deformation time series maps of descending InSAR. GNSS sites with displacement vectors are indicated by red arrows. The black ellipses represent the horizontal errors.

Since the processing of InSAR is based on a specific reference point to provide relative observations, and the results of GNSS are absolute measurements, there is a systematic bias between the two types of results. They cannot be directly fused. Based the method in Section 2.1, GNSS data were used to correct the InSAR ascending and descending results, respectively.

In order to evaluate the effect of the correction, as shown in Figure 8, six GNSS stations were selected to compare the results before and after the correction. We conducted a comparison between the InSAR and GNSS-derived displacement time series. The GNSS-derived 3-D displacement time series were projected into the LOS directions of the ascending and descending orbits, respectively. It can be seen from the figure that the InSAR observations showed a systematic deviation from the GNSS as a whole. Overall, the InSAR observations were smaller because they are relative observations based on reference points. When the InSAR observations were corrected using the polynomial fitting method, they were in good agreement with GNSS.

For quantitative analysis, the root mean square errors (RMSEs) of the difference between the pre- and post-correction InSAR observations and the GNSS value at the corresponding moments are marked in blue and red in the lower right corner of each subgraph. Taking the BGIS site as an example, the RMSE of the difference between the GNSS and the ascending orbit InSAR observation before the correction was calculated as 34.23 mm. This reduced to 3.82 mm after the correction. Therefore, the correction effect was noticeable. The correction results for the rest of the sites in Figure 8 also show that the overall difference between GNSS observations and corrected InSAR was smaller than that of uncorrected InSAR.

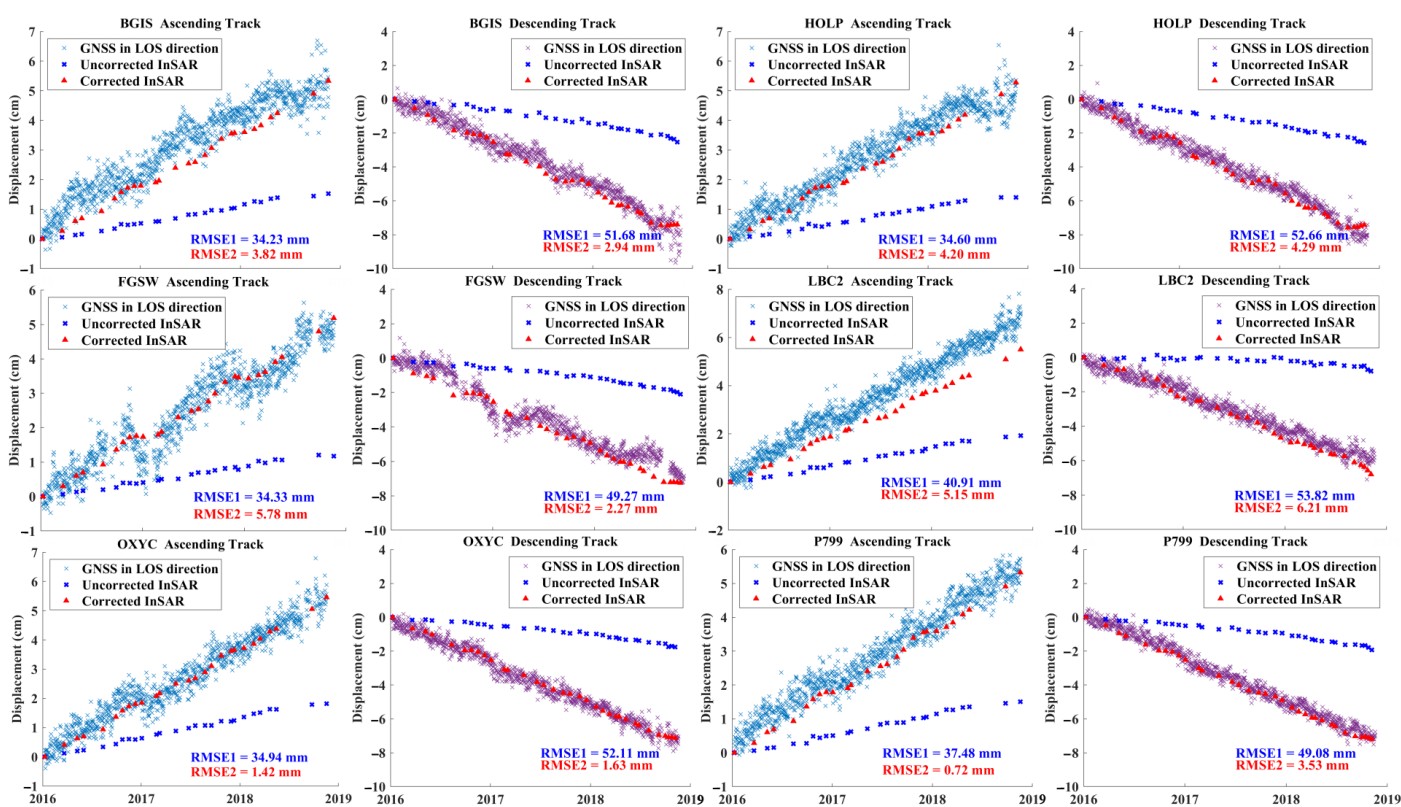

**Figure 8.** Comparison of the difference between GNSS and InSAR before and after the correction.

### 3.2. 3-D Deformation Time Series

Figures 9–11 show the eight deformation time series maps obtained using the SM-Kalman method in the east–west, north–south, and vertical directions, respectively. It was evident that Los Angles is characterized by rather uniform westward displacements (Figure 9) that accumulate over time. In addition, the Earth plate in the Los Angeles area moved northward as a whole and showed a trend of increasing from northeast to southwest. From the beginning of 2016 to the end of 2018, the maximum westward movement in the area reached 12 cm, and the northward movement reached 6 cm. The accumulated horizontal deformation over time is mainly due to the accumulation of interseismic stress on the San Andreas Fault [46].

In Los Angeles, the constant motion of the plate caused long-term deformation in the horizontal direction. In comparison, the displacement phenomena detected by the deformation time series in the vertical direction were basically caused by human activities. An uplift of approximately 18 mm was observed at Santa Fe Springs (near GNSS site SNTF) due to the swelling effect of oil production. Figure 11 also reveals that there was a prominent subsidence zone in the South El Monte area (near GNSS site WRNA), with a maximum deformation of approximately 12 mm.

Combined with the previous research results [47,48] and the current situation of urban development in this area, without considering the movement of the Earth, we speculate that, from 2016 to 2018, due to the exploitation of a large amount of groundwater, the city surface sank continuously and expanded to the surrounding area. The precipitation in Los Angeles was not abundant, and the rapid development of the city has made the water shortage in the area more serious [49]. In order to meet the needs of residents and industrial water, groundwater has been exploited in large quantities, resulting in long-term land subsidence in Los Angeles [50].

In addition, there are many faults in the Southern California area, making the region at high risk of earthquakes. This further restricts the mutual circulation of groundwater and makes it difficult to improve the phenomenon. A subsidence of approximately 18 mm was also found in Beverly Hills, northwest of GNSS station P800, due to groundwater extraction. These observed deformation positions or magnitudes are basically consistent with previous studies, fully demonstrating the effectiveness of the proposed method.

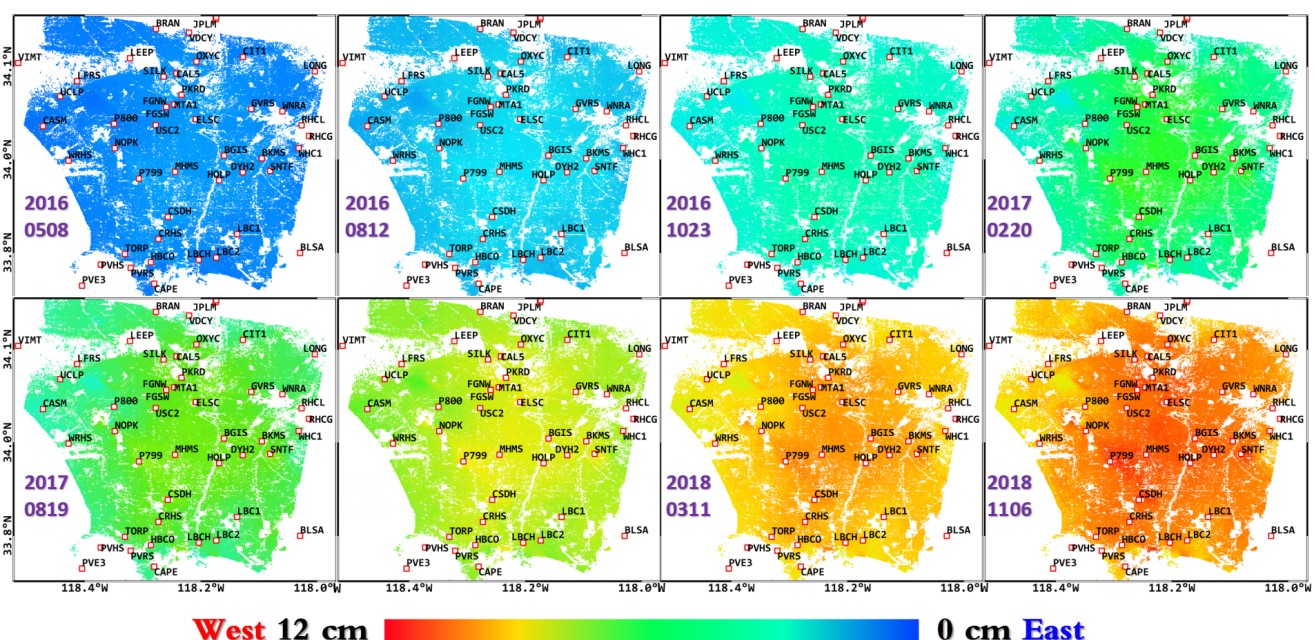

**Figure 9.** East–west deformation time series maps obtained by SM-Kalman. GNSS sites are represented by red boxes.

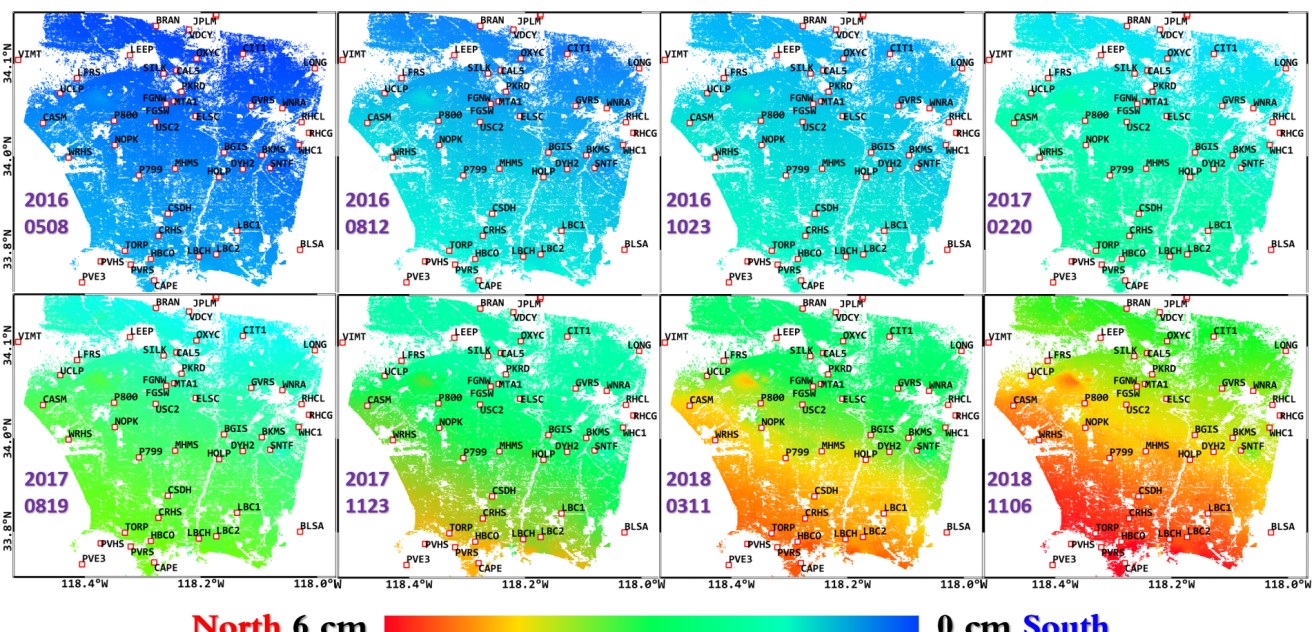

**Figure 10.** North–south deformation time series maps obtained by SM-Kalman. GNSS sites are represented by red boxes.

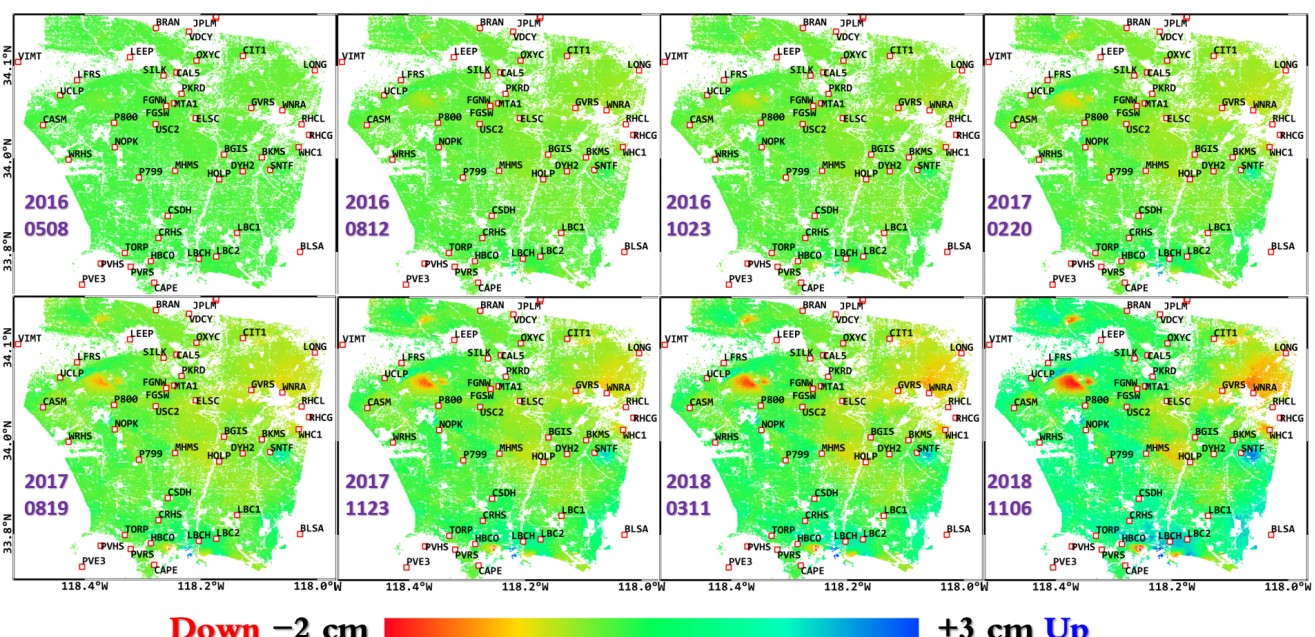

**Figure 11.** Up–down deformation time series maps obtained by SM-Kalman. GNSS sites are represented by red boxes.

In order to show the 3-D deformation time series more numerically, in Figure 12, the 3-D deformation time series of two types of time resolutions provided by the SM-Kalman method are displayed for four typical GNSS stations and are compared with the 3-D deformation time series of GNSS. The blue dots and red vertical bars with standard deviations identify the SM-Kalman results at GNSS time resolution (SM-kalman1) and InSAR time resolutions (SM-kalman2), respectively. The 51 moments of InSAR combine the observation moments of 27 ascending orbit data and 34 descending orbit data. Since the PSI results are only attached to discrete PS points, there is no guarantee that SM-Kalman-provided values exist at the location of every GNSS point.

For the GNSS points without the SM-Kalman filtering results, a $3 \times 3$ or larger window over each GNSS site was used to calculate the average displacements from the SM-Kalman filtering results.

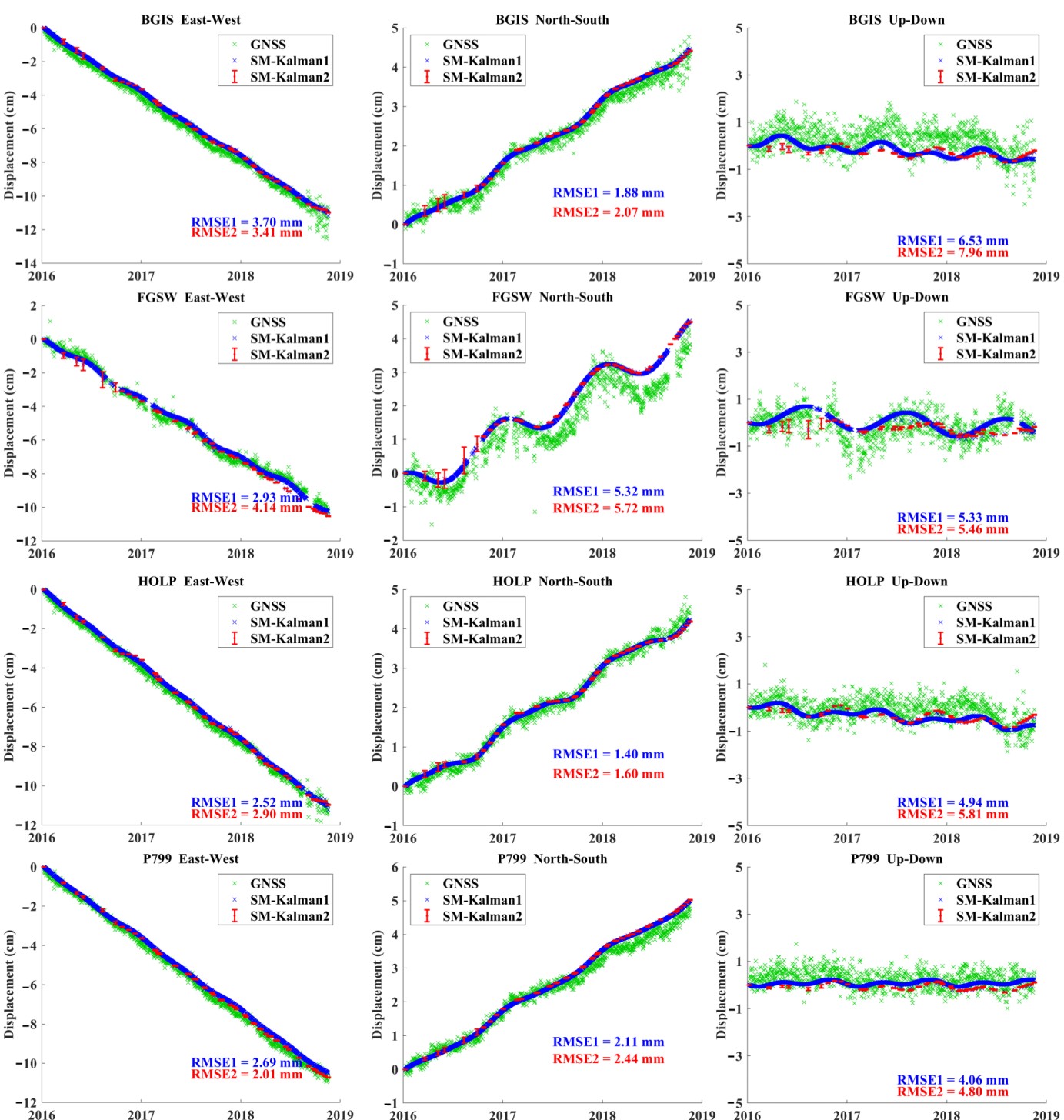

**Figure 12.** Comparison of GNSS results and deformation time series provided by SM-Kalman at two different temporal resolutions.

The 3-D displacements from $k = 0$ to $k = n - 1$ moments in Figure 12 were all obtained in the spatial domain using the SISTEM method. The SM-Kalman method was used in the space-time domain from the moment $k = n$. It can be seen from the red error bars that the

standard deviations of the SM-Kalman method were significantly lower than those of the SISTEM method.

This was expected because the inversion characteristics of the deformation were included in the time domain. The RMSEs between the SM-Kalman1, SM-Kalman2 methods and the GNSS results are also marked in blue and red and are shown at the bottom of each subgraph. Overall, the SM-Kalman1 results were in better agreement with GNSS, with smaller RMSEs. This shows that with the continuous increase of observation moments in the time domain, the accuracy of the time update will also be improved. This is not surprising, considering that the coefficients of the state equation were estimated using least squares.

The increase in redundancy increased the accuracy of the estimation. It is more evident that the results of SM-Kalman1 in the vertical direction better reflected seasonal changes than the results of SM-Kalman2, although the equations for both took into account semi-annual and one-year seasonal terms. As shown in Figure 12, the RMSEs of the SM-Kalman results were generally more significant in the vertical direction.

This is mainly attributed to two reasons: On the one hand, the vertical displacement in Los Angeles was affected by human factors and periodic terms, making the vertical deformation characteristics more complex. On the other hand, the results of GNSS had evident errors in the vertical direction, and it can be seen from Figure 12 that there were large fluctuations, which makes the RMSEs more significant when the GNSS results are compared with SM-Kalman as the true value.

## 4. Discussion

In contrast to the traditional analytical optimization method, SISTEM and other methods that only focus on the fusion of GNSS and InSAR data in the spatial dimension, the advantage of SM-Kalman is that it considers both the observation in the spatial dimension and the evolution characteristics of the deformation in the time dimension. The Kalman gain matrix **K** acts to balance them. As shown in Figure 12, the standard deviations of SM-Kalman were smaller than the results of SISTEM in the first few moments. If the time update step is not performed, SM-Kalman will degenerate into the SISTEM method, which only relies on the strain model to estimate the 3-D deformation. Another advantage of the SM-Kalman method is that the 3-D deformation can be predicted using the time update when there are no observations.

Figure 12 also reflects a feature of SM-Kalman in that the estimated 3-D deformation results in the north–south direction were more consistent with GNSS. This is intuitively reflected in the RMSEs between 3-D deformation and GNSS. Except for the strong seasonal deformation at the FGSW station, which increased the RMSEs, the RMSEs in the north–south direction of the rest of the stations were smaller than those in the east–west and vertical directions. This was determined using the observation geometry of the SAR satellite flying in the near north–south direction, which made the InSAR observations very insensitive to the deformation in the north–south direction.

In addition, due to the small incident angle of SAR satellites, InSAR is the most sensitive to the deformation in the vertical direction [12]. These facts led to the phenomenon in which the deformation results in the north–south direction were mainly contributed by GNSS. The flexibility of the SM-Kalman method is that the state equation can be established according to the actual situation of the deformation. In the Los Angeles area, the deformation exhibited strong seasonal variation, especially in the vertical direction.

Therefore, *T* was set to both 1 and 0.5. This can be determined via the pre-analysis of the GNSS deformation time series, such as the GNSS station FGSW in Figure 12. In Figure 13, the effects of three state equations based on different deformation fitting methods on the 3-D deformation time series are compared. Blue, red and green error bars indicate the results of three methods, namely, SM-Kalman (Polyfit) for cubic polynomial fitting, SM-Kalman (2PI) for fitting only one-year seasonal terms and SM-Kalman (2PI+4PI) for fitting one-year and half-yearly seasonal terms.

The RMSEs of the three different SM-Kalman methods compared to GNSS are shown below each figure. The results of SM-Kalman (2PI) and SM-Kalman (2PI+4PI) were very similar in three directions. Considering that in Figure 12, the results of SM-Kalman1 in the vertical direction better reflect seasonal changes than SM-Kalman2 (actually 2PI+4PI), a conclusion can be drawn. This is to say that only when the data involved in deformation fitting are extensive, for example, the estimated resolution of SM-Kalman1 is the same as that of GNSS, the small vertical semi-annual seasonal changes can be accurately captured.

This has little to do with whether there is a seasonal term with a semi-annual cycle in the state equation. SM-Kalman (2PI) is recommended when users pay little attention to very small seasonal changes in the vertical direction and are only willing to obtain 3-D deformation time series at the resolution of InSAR observations. This reduces the coefficients of fitting so that more accurate 3-D displacements can be estimated through SM-Kalman rather than the SISTEM method from an earlier moment. The results of SM-Kalman (Polyfit) in the east–west and vertical directions were comparable to those of the other two methods.

However, the errors in the north–south direction were more significant, especially at the FGSW station. The east–west deformation in the Los Angeles area had a linear trend to the west, that is, the low-frequency signal was dominant. Both the polynomial fitting and deformation model fitting methods can characterize this signal. As mentioned above, under the resolution of InSAR observation, neither SM-Kalman (2PI) nor SM-Kalman (2PI+4PI) could accurately reflect the small annual and semi-annual seasonal deformation. Figure 13 shows that this was also the case for SM-Kalman (Polyfit).

However, in the north–south direction, when there were strong seasonal and linear cumulative deformation trends, it was more difficult to obtain an accurate estimation using SM-Kalman (Polyfit), than with the other methods. In this case, increasing the polynomial order or replacing the fitting function by other methods, such as cubic splines and orthogonal polynomials, may improve the results. However, this is not discussed in detail here.

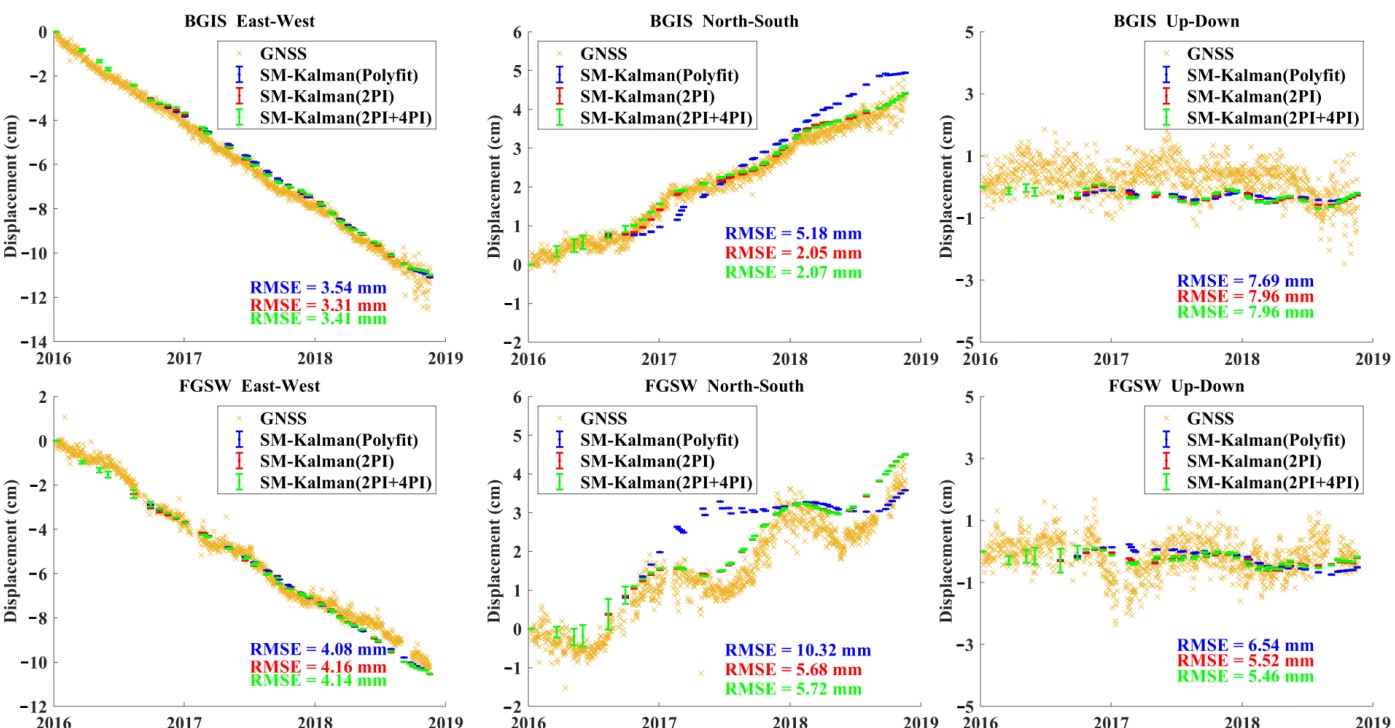

**Figure 13.** Comparison of the GNSS results with the deformation time series provided by SM-Kalman for three different fitting functions.

In order to further analyze the influence of the observation moments on the coefficients in the state equation, Figure 14 shows the changes in the five coefficients of SM-Kalman (2PI+4PI) with the increase in observation moments in the east–west, north–south and vertical directions, respectively.

The fitted coefficients oscillated in the first few moments due to the small number of observations; however, as the number of observation moments increased, the coefficients quickly became stable. $a_1$ mainly reflects the trend of linear deformation. The absolute value of $a_1$ in the east–west direction was the largest, followed by the north–south direction, and the vertical direction was close to 0. $a_2$ and $a_3$ reflect the seasonal terms with a one-year cycle. In the north–south direction and the vertical direction, $a_2$ and $a_3$ were the largest, and the east–west direction was close to 0, which is consistent with the 3-D deformation trend of the GNSS station BGIS in Figure 12.

In the east–west direction, the low-frequency signal was dominant. In the north–south direction, the low-frequency and high-frequency signals contribute together, and in the vertical direction, the high-frequency signal contributed the most. In addition, the absolute value of $a_5$ in the vertical direction was more significant than that in the other two directions. This again indicated a more complex semi-annual seasonal variation in the vertical direction, although the magnitude was minimal.

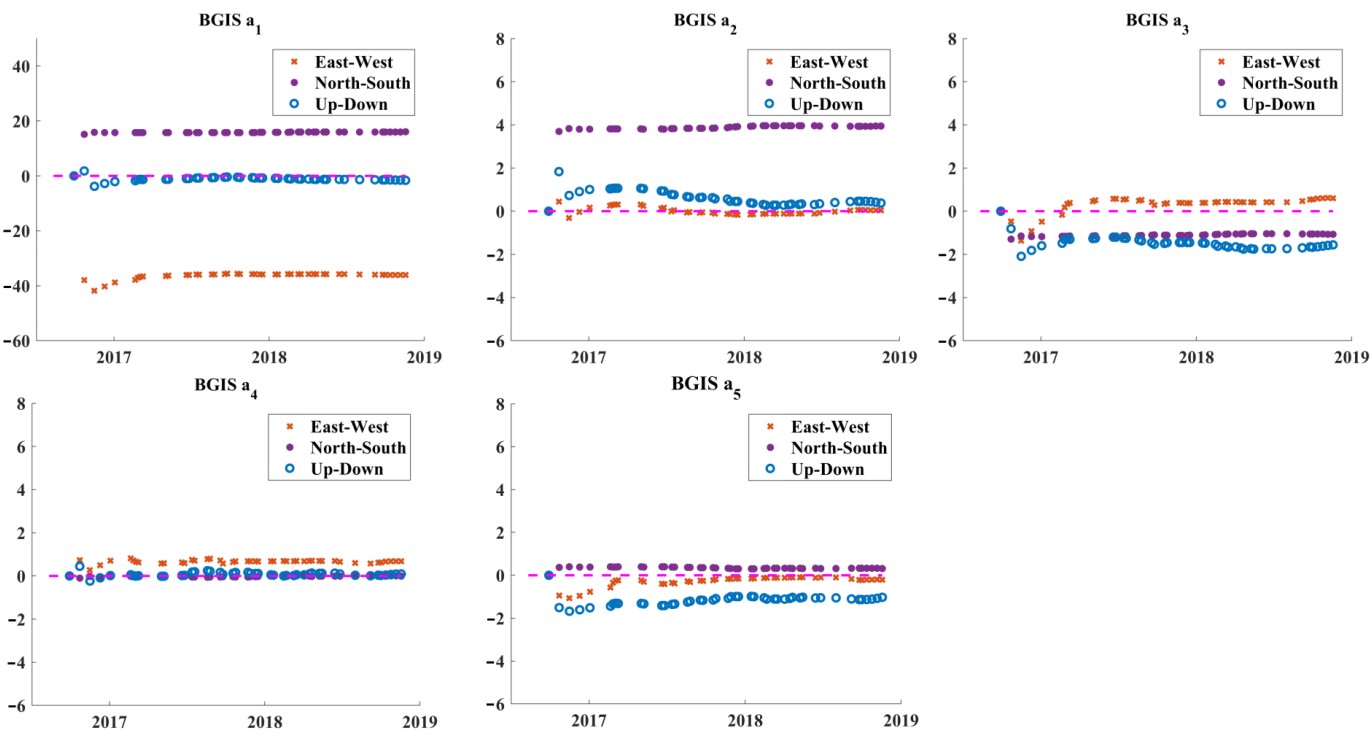

**Figure 14.** The five fitting coefficients of SM-Kalman (2PI+4PI) vary with the observation moments.

In this paper, the state equation of SM-Kalman was established based on the assumption that the deformation is smooth. This assumption is valid when no significant deformation discontinuities occur. However, in many cases, the deformation is discontinuous in both the time and space domains. The strain model has been demonstrated to estimate 3-D deformation fields for spatially discontinuous deformation regions, such as earthquakes [15] and volcanoes [16]. As for discontinuities in the time domain, SM-Kalman cannot be used directly since it assumes that the deformations are temporally correlated. We suggest two strategies to deal with this problem. One is that the SM-Kalman method can estimate the deformation before and after the event, respectively. Sharp deformation events can be easily distinguished using GNSS time series data.

The other is to add step terms that deal with co-seismic jumps to the fitting function of the deformation. In the Los Angeles case, 3-D deformation occurred continuously in both time and space. The SM-Kalman method should be further validated in more scenarios in the future.

## 5. Conclusions

In this paper, we demonstrated a new technique named SM-Kalman based on GNSS and InSAR data to obtain high-precision 3-D surface deformation time series. The method establishes the state equation by fitting the deformation time series of previous moments, which can effectively reflect the linear and seasonal terms of deformation. The observation equation is based on the strain model, which can utilize the observation data existing at each moment as much as possible, such as GNSS, ascending track InSAR and descending track InSAR.

Based on the established state equation and observation equation, the Kalman filter can dynamically estimate high-precision 3-D surface deformation time series from InSAR and GNSS observation data through the steps of time and measurement updates. Estimated 3-D deformation time series are beneficial in comprehensively monitoring the Earth's movement and surface deformation caused by human activities. SM-Kalman can estimate two types of 3-D deformation time series with the same temporal resolution as InSAR or GNSS observations according to users' needs. It can be seen that the new method achieves a high degree of fusion of GNSS and InSAR data in the time and space domains.

**Author Contributions:** Conceptualization, P.J.; methodology, P.J.; software, P.J.; validation, P.J. and X.L.; formal analysis, P.J.; investigation, P.J. and R.W.; resources, P.J. and R.W.; data curation, P.J.; writing—original draft preparation, P.J.; writing—review and editing, P.J., R.W. and X.L.; visualization, P.J.; supervision, P.J., R.W. and X.L.; project administration, X.L. and R.W.; funding acquisition, X.L. All authors have read and agreed to the published version of the manuscript.

**Funding:** This research was funded by the LuTan-1 L-Band Spaceborne Bistatic SAR Data Processing Program, grant number E0H2080702.

**Data Availability Statement:** The Sentinel-1 ascending and descending track SAR data for are available at https://www.esa.int/ESA (accessed on 7 June 2022). An European Space Agency account is required. The GNSS daily time series in the Los Angeles area are available on the Nevada Geodetic Laboratory website http://geodesy.unr.edu/gps_timeseries (accessed on 7 June 2022). The AW3D standard digital elevation model are available at https://www.eorc.jaxa.jp/ALOS/en/aw3d30/data/index.htm (accessed on 7 June 2022) via FTP upon submission of data access request.

**Acknowledgments:** The authors would like to thank the European Space Agency for providing Sentinel-1 data, the Nevada Geodetic Laboratory for providing GNSS time series and Japan Aerospace Exploration Agency for providing AW3D standard digital elevation model in Los Angeles.

**Conflicts of Interest:** The authors declare no conflict of interest.

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
