# Peer review of "Deriving 3-D Surface Deformation Time Series with Strain Model and Kalman Filter from GNSS and InSAR Data"

_remotesensing, doi:10.3390/rs14122816_

Round 1
Reviewer 1 Report
Major comments
- The authors have done a good job of laying out the approach of the fusion between GNSS information and InSAR data for deformation monitoring. The methodology was laid out well, easy to follow and repeatable which is ideal for other contributions to build on it.
- The entire manuscript needs to be thoroughly proof-read. Though it is well written, there are many incoherent phrases and sentences which are hard to understand. An attempt is made by reviewer to highlight some of these in the specific comments below but authors need to parse through the document fixing many more of this.
- Concerning the GNSS observations, how did the authors account for the errors associated with the line of sight, for instance, ionospheric and tropospheric refractions, Sagnac effects, etc? Can these errors be ignored all together? If so, why and if not, can the authors comment on this further?
- References: Most of the references in this paper are missing page numbers. Please fix these and ensure proper references.
General comments:
Line 28: "The existing continuous GNSS measurement can provide high-precision (millimeter level)..." This is a very optimistic statement that GNSS measurements can provide millimetre precision. This can only happen using enhanced processing techniques. Ideally, centimetre level of accuracy is possible with high-precision processing. Please provide references.
Line 40: "many scholars have conducted a lot of researches on the combine of ..." Authors meant to say "combination"? Please rephrase.
Lines 94-96: Sentence is hard to read. Please rephrase
Figure 5: What does the author mean by GNSS here? These seem like GNSS receiver stations and not GNSS constellations. Please use the appropriate terms correctly
Line 308: "Multi-looking operation with 2 looks in the range..." What does this "2" refer to?
Line 313: "...employed to remove the topographic effect..." What is this topographic effect? No prior explanation has been offered before using this term? Authors should expand more on this effect
Author Response
Dear Reviewer,
We would like to thank you very much for your valuable comments and good suggestions that greatly helped to improve our manuscript.
We have carefully considered your valuable comments and good suggestions. Please see the attachment for responses to all comments.
With best wishes
Yours sincerely
Corresponding author:
Name: Xiaolei Lv
E-mail: academism2017@sina.com

Reviewer 2 Report
In the beginning of the manuscript there are some language issues, however it does improve. I do appreciate the work however in testing a new theory, the study area is very complex and full of variables that complicate a simple demonstration of a technique.
Author Response
Dear Reviewer,
We would like to thank you very much for your valuable comments to improve our manuscript.
The manuscript has been carefully revised. The amendments are all highlighted in the revised manuscript.
With best wishes
Yours sincerely
Corresponding author:
Name: Xiaolei Lv
E-mail: academism2017@sina.com
Reviewer 3 Report
In thsi study SM-Kalman method proposed for deformations. the newly proposed SM-Kalman method can not only produce high-precision 15 deformation results at the millimeter level, but also provide 3-D deformation time series with two 16 time resolutions of InSAR or GNSS according to the needs of users. The new method achieves a 17 high degree of temporal and spatial fusion of GNSS and InSAR data.
I can recommend some paper to improve the quaity of introduction
Demirci, Åž. & Özdemir, C. (2021). An investigation of the performances of polarimetric target decompositions using GB-SAR imaging . International Journal of Engineering and Geosciences , 6 (1) , 9-19 . DOI: 10.26833/ijeg.665175
Konak, H. , Küreç Nehbit, P. , Karaöz, A. & Cerit, F. (2020). INTERPRETING DEFORMATION RESULTS OF GEODETIC NETWORK POINTS USING THE STRAIN MODELS BASED ON DIFFERENT ESTIMATION METHODS . International Journal of Engineering and Geosciences , 5 (1) , 49-59 . DOI: 10.26833/ijeg.581584
Duysak, H. & YiÄŸit, E. (2022). Investigation of the performance of different wavelet-based fusions of SAR and optical images using Sentinel-1 and Sentinel-2 datasets . International Journal of Engineering and Geosciences , 7 (1) , 81-90 . DOI: 10.26833/ijeg.882589
YiÄŸit, E. , Demirci, Åž. & Özdemir, C. (2022). Clutter removal in millimeter wave GB-SAR images using OTSU’s thresholding method . International Journal of Engineering and Geosciences , 7 (1) , 43-48 . DOI: 10.26833/ijeg.867467
proposed a new technique named SM-Kalman based on GNSS and InSAR data from temporal and spatial dimensions to obtain high-precision 3-D surface deformation time series has been accepted after minor corroctions.Pleasde improve introduction section
Author Response
Dear Reviewer,
We would like to thank you very much for your valuable comments and good suggestions that greatly helped to improve our manuscript.
We have carefully considered your valuable comments and suggestions. In the newly submitted manuscript, we have improved the Introduction section. All changes are highlighted in yellow.
With best wishes
Yours sincerely
Corresponding author:
Name: Xiaolei Lv
E-mail: academism2017@sina.com
Round 2
Reviewer 2 Report
thank you for your quick response and edits.
Author Response
Dear Reviewer,
We would like to thank you very much for your valuable comments and good suggestions that greatly helped to improve our manuscript.
We have thoroughly revised the grammar and sentences of the manuscript, through MDPI's English editing services. Moreover, the formulas, figures and tables in the manuscript have been thoroughly checked. The amendments are all highlighted in the revised Latex manuscript.